# Facilitation of hERG Activation by Its Blocker: A Mechanism to Reduce Drug-Induced Proarrhythmic Risk

**DOI:** 10.3390/ijms242216261

**Published:** 2023-11-13

**Authors:** Kazuharu Furutani

**Affiliations:** Department of Pharmacology, Faculty of Pharmaceutical Sciences, Tokushima Bunri University, 180 Nishihama-Boji, Yamashiro-cho, Tokushima 770-8514, Japan; furutani@ph.bunri-u.ac.jp

**Keywords:** hERG channel, *I*
_Kr_, drug-induced arrhythmias, molecular pharmacology, structural model

## Abstract

Modulation of the human Ether-à-go-go-Related Gene (hERG) channel, a crucial voltage-gated potassium channel in the repolarization of action potentials in ventricular myocytes of the heart, has significant implications on cardiac electrophysiology and can be either antiarrhythmic or proarrhythmic. For example, hERG channel blockade is a leading cause of long QT syndrome and potentially life-threatening arrhythmias, such as *torsades de pointes*. Conversely, hERG channel blockade is the mechanism of action of Class III antiarrhythmic agents in terminating ventricular tachycardia and fibrillation. In recent years, it has been recognized that less proarrhythmic hERG blockers with clinical potential or Class III antiarrhythmic agents exhibit, in addition to their hERG-blocking activity, a second action that facilitates the voltage-dependent activation of the hERG channel. This facilitation is believed to reduce the proarrhythmic potential by supporting the final repolarizing of action potentials. This review covers the pharmacological characteristics of hERG blockers/facilitators, the molecular mechanisms underlying facilitation, and their clinical significance, as well as unresolved issues and requirements for research in the fields of ion channel pharmacology and drug-induced arrhythmias.

## 1. Introduction

The human *Ether-à-go-go-Related Gene* (hERG or KCNH2) encodes the alpha subunit of a time- and voltage-dependent potassium channel [1,2,3,4]. Its function is best understood in the heart, where it plays a crucial role in the robust repolarization of the cardiac action potential by mediating the *I*_Kr_ current [1,2,3,4,5,6]. The pharmacological blockade of hERG channels results in reduced *I*_Kr_ currents in ventricular myocytes, leading to delayed repolarization and prolonged action potential duration. This manifests as QT interval prolongation on electrocardiography, which is known as drug-induced long QT syndrome (LQT), potentially increasing the risk of life-threatening ventricular fibrillation and tachycardia, such as *torsades de pointes* [1,2,7,8,9,10,11,12,13].

Drug-induced LQT results from off-target inhibition of the hERG channel by structurally varied cardiac and non-cardiac drugs, and can pose a high risk of life-threatening arrhythmias [1,4,12]. Several drugs, including terfenadine, astemizole (antihistamine), sertindole (antipsychotics), grepafloxacin (antibacterial), cisapride (gastroprokinetic agents), and levomethadyl (opioid), have been withdrawn from the market due to this problem. However, assessing the risk of drug-induced lethal arrhythmias in humans before drug approval is ethically difficult. Therefore, the in vitro inhibition of hERG channels by drugs in cell cultures is considered a surrogate marker for drug-induced arrhythmias. In accordance with the drug safety guidelines developed by the International Council for Harmonization of Technical Requirements for Pharmaceuticals for Human Use (ICH), ICH S7B [14], all pharmaceutical companies currently conduct in vitro testing, known as the hERG test, to determine whether their drug candidates exhibit inhibitory activity on hERG channels. If a drug inhibits the hERG channels, it carries the risk of proarrhythmia. Notably, since the release of the ICH S7B guidelines in 2005, no approved drugs have been withdrawn from the market owing to hazardous arrhythmia risks, indicating that early testing of hERG channel block is extremely effective in eliminating the risk of approving potentially torsadogenic drugs.

However, the hERG test is an inadequate marker of true proarrhythmic risk, as several studies have demonstrated low proarrhythmic proclivities for hERG blockers [15,16,17,18]. Additionally, the hERG block in cell culture does not always correlate with arrhythmogenicity in humans, which may be attributed to the difference between the free therapeutic plasma concentration of drugs and the concentration of the hERG block [19]. Even if a drug exhibits hERG channel inhibitory activity, it may not pose a clinical problem if inhibition does not occur at the concentrations used for the treatment. Another reason is the absence of action potential duration or QT interval prolongation, which is typically expected with hERG block. This may be due to other effects of the drug, particularly on cardiac ion channels or transporters [20,21]. A good example is when a drug inhibits both hERG and calcium channels, resulting in no prolongation of the action potential. Considering these factors, the hERG test is not sufficiently selective for the assessment of life-threatening arrhythmias and may produce false positives [15,16].

Another concern is the high sensitivity of the hERG test. Reports indicate that 50–70% of the structurally diverse drugs tested exhibit some degree of hERG inhibitory activity [22,23,24]. The high sensitivity of the hERG test poses a challenge during drug development, as it complicates risk assessment.

Consequently, the identification of hazardous drugs that block the hERG channel remains an ongoing challenge and requires continued effort [15,16]. Addressing this challenge has the potential to enhance drug development efficiency, remove significant impediments to the approval of new drugs, and may incentivize the development and clinical application of safe hERG blockers for various diseases. Reports have also suggested the potential therapeutic utilization of hERG channel blockade or chaperone modulation in the treatment of cancer [25,26,27,28,29] and mental disorders [30], respectively, further underscoring the importance of improving our understanding of the function of this channel.

hERG blockers, such as amiodarone and nifekalant, are Class III antiarrhythmic agents that are used in clinical practice to terminate ventricular tachycardia and fibrillation [31,32,33,34,35]. These drugs are considered a last resort for patients who are unresponsive to other antiarrhythmic drugs; despite their proarrhythmic risks, their effectiveness and importance in clinical practice outweigh these risks. Intense pharmacological experiments have been conducted to understand the efficacy and safety of Class III antiarrhythmic agents. Notably, these studies have revealed that their effects on hERG channels are not solely blocking [31,32,33,34,35,36,37], but instead, they can facilitate the voltage-dependent activation of hERG channels, thus enhancing their activity near the membrane potential threshold for channel activation [36,37,38]. This increase in hERG current is achieved through a mechanism distinct from that of hERG channel activators [39,40,41,42]. This agonistic effect of hERG blockers, known as facilitation, has been observed not only in Class III antiarrhythmic agents but also in various non-cardiac drugs with hERG channel inhibitory activity used in clinical practice [43]. Importantly, recent studies have suggested that this drug action contributes to reducing the proarrhythmic risk associated with hERG blockers [44].

This review article seeks to offer a thorough examination of the present knowledge regarding the effects and underlying mechanisms of hERG channel facilitation by “blockers”, and to summarize the consequences of these effects on cardiac arrhythmias.

## 2. Facilitation of hERG Activation by Its Blocker

In response to membrane depolarization, hERG channels undergo slow activation followed by much more rapid inactivation [1,4], resulting in inward rectification in the current–voltage (IV) relationship with a maximal outward current at voltages between −10 and 0 mV. This gating property of hERG channels also results in a decreasing current with further depolarization, e.g., during the plateau phase of the ventricular myocyte action potential. The maintenance of this plateau is crucial for ensuring sufficient time for the entry of extracellular Ca^2+^ into the myocyte and Ca^2+^ release from the sarcoplasmic reticulum to enable cardiac contraction [1,4]. As the myocyte repolarizes, hERG channel conductance increases due to recovering from inactivation and deactivation. The *I*_Kr_ current during phase 3 repolarization of the ventricular action potential accelerates the repolarization and terminates the action potential. In experimental studies, the sigmoidal conductance–voltage (GV) relationship was analyzed by measuring tail currents at a negative voltage, where inactivation was weak. Many drugs interact with hERG channels and influence cardiac electrophysiology. Some drugs, known as blockers, reduce both the IV and GV relationships. Additionally, some hERG blockers not only reduce the IV and GV relationships but also shift them to the left, as shown in Figure 1 [36,37,38,43,45,46,47]. This leftward shift in the GV relationship is referred to as the “facilitation” of voltage-dependent activation by the drug. In cases where the leftward shift was significant, the IV relationships of the control and drug-treated conditions may intersect, resulting in an increase in drug-induced hERG currents from the control at membrane voltages near the activation threshold. However, hERG channel activators increase hERG channel currents through a mechanism different from that of hERG blockers/facilitators (referred to as “hERG facilitators” in this review). Specifically, known activators enhance hERG channel activity by inhibiting inactivation [39,41,48,49,50], whereas the mechanism of hERG facilitation, which does not affect the inactivation process, differs from that of its activation. Perry had effectively summarized the pharmacological differences between the drugs in their review article [42].

Numerous reports on hERG facilitators have been published [36,37,38,43,45,46,47]. Among these hERG facilitators, our group focused on Class III antiarrhythmic agents such as amiodarone and nifekalant [36,37,44,51]. These agents are highly effective in terminating refractory ventricular tachycardia and fibrillation and can induce voltage-dependent facilitation, as shown in Figure 1. After treatment with nifekalant, the hERG currents induced by the test pulses decreased at −50 mV, which is considered a block and is thought to be the antiarrhythmic mechanism of Class III antiarrhythmic agents by prolonging the action potential duration and relative refractory period of ventricular myocytes. However, when strong depolarization to +60 mV was applied, the subsequent test pulse to −50 mV induced a large hERG current. This large hERG current was only observed in the presence of nifekalant, indicating that the drug (nifekalant) is necessary to increase the current. As shown in Figure 1A_i_, the effect of a single strong depolarization was transient. The response to the test pulses gradually decreased and eventually returned to the response observed before the application of strong depolarization. These transient changes in hERG currents were caused by a leftward shift in the GV relationship (Figure 1B). A detailed biophysical assessment of this drug-induced leftward shift of the GV relationship revealed that both block and facilitation occurred with similar concentration dependence (Figure 2). Additionally, the extent to which a drug shifts its GV relationship to the left is inherent in the drug itself. Notably, amiodarone shifted the curve by approximately 30 mV, whereas nifekalant shifted it by approximately 25 mV to the left (Figure 2C). Although experiments are usually conducted using cells solely expressing hERG channels, a shift in the GV relationship also occurs when co-expressing the auxiliary subunit KCNE1 [44,52]. The expression system used did not significantly affect facilitation, even when recordings were carried out at different extracellular K^+^ concentrations. Unlike activators, facilitators had little effect on inactivation.

Certain voltage-dependent properties of the drug-induced facilitation aid the investigation of the mechanisms by which a drug can exert both hERG block and facilitation. Recently, the mechanism by which depolarization induces hERG facilitation was revealed. The voltage dependency of the induction of facilitation is associated with the voltage dependency of the hERG channel activation, specifically the opening of the activation gate in the pore [51]. In this study, the D540K hERG mutant was utilized, which can be activated by both depolarization and hyperpolarization [53]. In the wild-type hERG channel, facilitation is induced only by depolarization, whereas in the D540K hERG channel, it is induced by both depolarization and hyperpolarization stimuli [51]. Furthermore, this study demonstrated that drugs can facilitate activation through hyperpolarization in the D540K hERG mutant [51]. While the GV relationship of depolarization-induced activation shifted leftward, the GV relationship of hyperpolarization-induced activation shifted rightward. Considering the difference in structural changes in the voltage sensor domain caused by membrane depolarization and hyperpolarization, it is anticipated that drugs affect structural changes in the pore domain, facilitating the opening of the hERG channel pore when structural changes occur in the voltage sensor.

Nifekalant and other facilitators act as open-channel blockers for hERG. It is important to note that despite this, some readers may still question why depolarization (channel opening) only affects facilitation and not inhibition. The experimental protocol illustrated in Figure 1A utilizes 4 s long test pulses that are commonly used to stimulate slow-activating hERG channels. While the block also requires pore opening, this is not evident in the steady state following the channel opening. Furthermore, the blocking effect of the drugs was assessed by comparing the magnitude of inhibition with the current in the absence of drugs, which essentially evaluates the effect of the drug solely on the opened channels. This evaluation method makes it challenging to discern differences in inhibition at different membrane potentials. Conversely, the facilitation induced by the pre-pulse was evaluated, and the subsequent reopening by the test pulse did not typically cause significant pore opening. This double-pulse protocol enables clear observation of the voltage-dependent induction of facilitation.

## 3. Structural Basis of hERG Facilitation

These studies have fielded our current understanding of the structural basis of facilitation. As previously mentioned, the facilitation effect is not limited to antiarrhythmic drugs, but can also be observed with non-cardiac hERG-blocking drugs [43]. The structural basis of hERG facilitators has been studied in detail. A three-dimensional quantitative structure–activity relationship (3D-QSAR) model was developed using EC50 values for facilitation and drug structures, consisting of one positively ionizable feature and three hydrophobic features [43]. The 3D-QSAR model was experimentally validated using test compounds, confirming the existence of common mechanisms among drugs for facilitation. Although this pharmacophore model of facilitation shares the same types and number of features as the block pharmacophore, their spatial arrangements differ slightly [43,54]. From this comparison, it was observed that there are drugs that satisfy both pharmacophore models for hERG channel block and facilitation, but the interaction with the channel may differ between block and facilitation.

The initial structure of the hERG channel was provided by the MacKinnon lab using single-particle cryo-electron microscopy [55]. The voltage sensor in this hERG structure had a non-domain-swapped architecture and appeared to be in a depolarized, activated conformation. The pores of the structure apparently opened. Subsequently, Asai et al. reported the structure of hERG complexed with astemizole, a representative hERG-blocking drug [56]. This structure clearly demonstrates that the drug interacts with several amino acids, including S624 and Y652 on the S6 helix, forming the central cavity of the hERG pore, thus supporting previous studies [1,57,58,59,60,61,62,63,64,65,66]. The involvement of F656 in direct interactions remains unclear, and it may play a role in mediating the access of the drug to the binding site [56,67]. Notably, astemizole is a classical hERG blocker. Therefore, these structures do not reveal the structural basis of the facilitation.

Hosaka et al. conducted alanine scanning of the pore and S6 helices of the hERG channel and analyzed the impact of mutations on both blockade and facilitation [36]. This study revealed that several mutants failed to exhibit both block and facilitation. For instance, substitutions of Y652 or F656 with alanine, which have been known to be crucial residues for the block, abolished the facilitation by nifekalant. Y652 is also crucial for the facilitation of phenanthrene [47]. These findings suggest that facilitation, similar to the block, occurs through interactions within residues that form the central cavity of the hERG pore. Importantly, mutants that had no effect on the block but affected facilitation were also identified. Some mutants, such as S624A, L646A, M651A, S654A, G657A, V659A, and L666A, inhibited facilitation, whereas others, such as I647A, S649A, S660A, I663A, R665A, and Y667A, enhanced it. It appears that residues around the central cavity of the hERG are specifically involved in facilitation and play a distinct role from that of the block. Structural models of open- and closed-state hERG channels were built to illustrate the amino acid residues associated with facilitation and putative drug interaction (Figure 3) [36]. We conducted a study using Rosetta to investigate hERG channel–drug interactions and found that certain drugs, known as hERG facilitators, are positioned in the hydrophobic pocket of the hERG central cavity [68]. This pocket is thought to play a role in drug interactions and is located in the open-state hERG channels, as identified by Wang and MacKinnon [55]. Our modeling study also suggested four potential fenestration regions in the pore, but hERG facilitators, including nifekalant and amiodarone, were not positioned in these fenestration regions [68]. In combination, these findings suggest that the hydrophobic pocket may be involved in the facilitation of hERG channels by these drugs. This intriguing possibility requires further validation and study.

Our working hypothesis for drug-induced facilitation (Figure 4) can be summarized as follows. Drugs enter the channel and inhibit channel function when the activation gate of the channel opens due to depolarization. It is then possible that drugs become trapped within the channel when the channel closes [69,70,71,72]. Drug interactions within the hydrophobic pocket may affect the closure of the S6 helix bundle, thereby altering the channel gating. Consistent with our results, experimental data suggest that facilitating drugs may act as a wedge to bias the hERG channel gating equilibrium toward an open state conformation. This can increase the hERG current amplitude in response to low-voltage depolarization [37,51]. Additional experimental measurements and molecular dynamics simulations are required to test structural model-based hypotheses.

## 4. A Possible Role of hERG Facilitation in hERG Block-Associated Arrhythmia

Yamakawa et al. investigated the facilitation effects of various non-cardiac drugs that block hERG channels [43]. The results revealed that drugs such as fluoxetine (an antidepressant), haloperidol (an antipsychotic), and chlorpheniramine (an antihistamine) exhibit both conventional blocking effects on hERG channels as well as facilitation effects similar to antiarrhythmic agents, such as amiodarone and nifekalant. This suggests that the facilitation effect is not exclusive to antiarrhythmic agents and may be commonly observed among clinically used hERG blockers. This study also reported that some classical hERG blockers, including atenolol, terfenadine, and sotalol, did not exhibit facilitation [43]. Terfenadine is an antihistamine that has been withdrawn from the market owing to its high risk of life-threatening arrhythmias. These findings suggest that the facilitation effect of hERG blockers may help reduce the risk of arrhythmias through their hERG-blocking mechanism. However, assessing the occurrence of facilitation in vivo presents technical challenges. In terms of the drug’s concentration, nifekalant can exert block and facilitate hERG channels in almost the same concentration-dependent manner [37]. In addition, the repetitive excitation of ventricular myocytes may trigger facilitation. Experimental evidence has demonstrated that repeated stimulation with a voltage-clamp waveform that resembles action potentials can induce maximal facilitation [44]. This may suggest that facilitation occurs in living hearts.

An in vitro validation was performed to examine the effects of nifekalant, a hERG blocker with facilitation, and dofetilide, a hERG blocker without facilitation, on rat ventricular myocyte action potentials. Even at concentrations that inhibited *I*_Kr_ to a similar extent, dofetilide was more likely to induce early afterdepolarizations (EADs) than nifekalant, as shown in Figure 5A,B [44]. This suggests that facilitation may reduce the risk of arrhythmia. To further explore this concept, a theoretical study was conducted. We developed a mathematical model to simulate facilitation and its impact on action potential waveforms [44]. The facilitation model was formulated as a drug-induced shift in the GV relationship. By incorporating hERG block and facilitation models into a human ventricular myocyte action potential model (ORd human ventricular AP model [73]), we examined the influence of facilitation. Without a facilitation mechanism, increasing the concentration of a classical hERG blocker resulted in action potential prolongation and the development of EADs (Figure 5C,D). However, in the presence of a facilitation mechanism, the prolongation of the action potential was suppressed, and a higher concentration was required to induce EADs (Figure 5C,D). These observations theoretically illustrate that facilitation can lower the proarrhythmic risk associated with a drug.

Class III antiarrhythmic agents are hERG blockers used clinically to suppress ventricular tachyarrhythmias [1,31,32]. To suppress tachyarrhythmias without provoking *torsades de pointes*, the *I*_Kr_ block would ideally be use-dependent and prolong APD only in response to high-frequency stimulation [7,74]. However, reverse frequency-dependent action on APs is a property common to Class III antiarrhythmic agents [33,45,46,74,75,76,77], and the associated proarrhythmic risk limits their clinical usefulness [74,78]. The reverse frequency dependence of *I*_Kr_ block was first explained by an increase in the slowly activated delayed-rectifier K^+^ current with rapid heart rate [75]. Thus, we theoretically tested whether the presence or absence of facilitation also influences the reverse frequency-dependent effect and suggested that facilitation reduces this dependence [44]. These effects are particularly prominent in models of heart failure [44], in which functional remodeling predisposes patients to arrhythmias [79].

The mechanisms underlying these effects have also been analyzed [44]. It is important to recall that the presence of blockers, such as nifekalant, can result in the crossing of the GV and IV relationships compared to their absence (Figure 1 and Figure 2) [36,37]. When considering the trajectory of the ventricular action potential, depolarization occurs during phase 0 of the action potential, and this moves the myocyte membrane potential through the range of membrane potentials where facilitation increases the *I*_Kr_/hERG current. Consequently, the effects of facilitation were minimal during phases 0 and 2. As the action potential repolarization begins and the myocyte voltage returns to the range of membrane potentials where facilitation occurs, facilitation increases the *I*_Kr_/hERG currents. As a consequence, in this time- and voltage-window, the repolarizing currents are larger than in the absence of the drug. During a single ventricular action potential, hERG channel facilitators change their attributes to *I*_Kr_. It decreases the current first and then increases it later, which prevents excessive prolongation of the action potential and repolarization impairments (Figure 6).

This facilitation-induced increase in the *I*_Kr_/hERG current during phase 3 can become even more pronounced when repolarization is delayed [44]. This can be explained by the fact that, as the action potential duration is prolonged, the duration of the membrane potential within the range where facilitation occurs is also prolonged. This property can significantly affect the concentration-dependent and reverse frequency-dependent effects of hERG-blocking drugs on action potential duration. At low concentrations of these drugs, which only slightly prolong the action potential, the facilitation effect has a minimal impact. However, at higher concentrations, the action potential duration is further prolonged and the impact of facilitation can become more significant. Similarly, at high excitation frequencies, where the action potential is relatively short, the facilitation effect is less influential. However, at low excitation frequencies, where the action potential is relatively longer, it becomes more susceptible to facilitation. This mechanism allows the facilitation effect to prevent further prolongation of the action potential when it has already been extended by a drug-induced blockade.

Class III antiarrhythmic drugs are believed to exert their antiarrhythmic effects by extending the refractory period through hERG blockade, thereby prolonging the duration of action potential. There is some concern that the facilitation effect of hERG blockade on *I*_Kr_ current might also affect the antiarrhythmic effects of Class III antiarrhythmics. However, simulations have indicated that this impact is minimal. The increase in *I*_Kr_ current during phase 3 reduces excitability, offsetting the shortening effect on action potential duration. In simulations, prolongation of the relative refractory period by a hERG blocker with a facilitation effect was comparable to that of a classical hERG blocker [44].

Additionally, there is no report demonstrating that the facilitation effect in other Kv channels, such as *I*_Ks_ channels, contributes to repolarization in myocytes.

Further experimental validation is required; however, the notion that the proarrhythmic risk of hERG-blocking drugs may be mitigated by the facilitation effect is worth pursuing. In addition, to fully evaluate the risks associated with hERG-blocking drugs accurately, the presence or absence of facilitation effects must be considered.

## 5. Conclusions and Future Direction

In this review, we present an overview of the properties of hERG channel facilitation, its critical role in pharmacotherapy, and the current understanding of its functional and structural aspects. This material serves as the initial basis for the mechanism by which drugs facilitate hERG channel activation and suggests its clinical implications.

Important remaining questions include the following.

(i) How do drugs facilitate hERG activation? Which process(es) of hERG opening/closing are affected by the drug? It is highly likely that the drug is located within the central cavity of the hERG pore when it exerts facilitation [36,47,51]. By further investigating hERG channel gating and modulation, we can acquire a more profound understanding of its behavior. Advanced methods, such as single-channel recording and (single) molecular imaging, can be used to detect precise modifications in the hERG gating process triggered by the drug.

(ii) Where do the drugs bind exactly? To ascertain the structural location of the drug at the binding site for hERG activation, is it the hydrophobic pocket, or are other sites responsible for facilitation? Additionally, can the binding sites vary in size or shape between the closed and open states? If so, this may explain why certain drugs are capable of facilitating channel activation. Furthermore, it is conceivable that drugs within the channel pore can obstruct or eliminate hERG channel inactivation. Molecular dynamics simulations [80,81,82,83,84,85,86] would be particularly useful in revealing any modifications in the binding site resulting from drug binding in both closed and open states.

(iii) What are the structural differences between a hERG blocker/facilitator and a classical hERG blocker? A wide range of structurally diverse compounds interact with the hERG channel and inhibit its function. Is it possible to predict the facilitation effects of drugs based on their structures without experimental data? Although previous attempts have been made, the predictive capability of these methods has been limited to only a few selected drugs [43]. The facilitation effect, characterized by a leftward shift in the GV relationship, is unique to each drug. As such, the structure of the drug is likely related not only to its affinity but also to the strength of facilitation (modulation of hERG activation). Detailed exploration of pure hERG facilitators has not yet been conducted, and further research is necessary to understand the structural basis of the facilitation effect of drugs. Ultimately, this knowledge is crucial for the rational control and design of the facilitation effect of drugs.

(iv) How can we apply our understanding of the hERG facilitation to advance drug discovery and treatment? In the field of medical healthcare, improving the effectiveness and safety of pharmacotherapy remains an ongoing goal, requiring collaboration between industry, government, and academia. The hERG channel has garnered significant attention because of its association with arrhythmias, and an integrated approach incorporating multiscale, multiphysics biological simulations is essential to comprehensively understand and predict complex phenomena such as cardiac arrhythmias. To advance drug discovery and treatment, further testing of the hypothesis that “facilitation reduces the proarrhythmic risk of hERG blockers” [44] through theoretical research and proof-of-concept experiments is crucial. In principle, this knowledge could then be utilized to inform decision-making regarding drug therapy. Understanding the structure–activity relationship of drugs also opens up possibilities for theoretical drug design and modification, contributing to drug discovery. Although caution must continue to be exerted in the targeting of the hERG channel in drug discovery, if safety is confirmed, the conventional belief that “hERG blockers are dangerous” may evolve, possibly allowing vast resources of hERG-active compounds to be utilized for drug development.

## Figures and Tables

**Figure 1 ijms-24-16261-f001:**
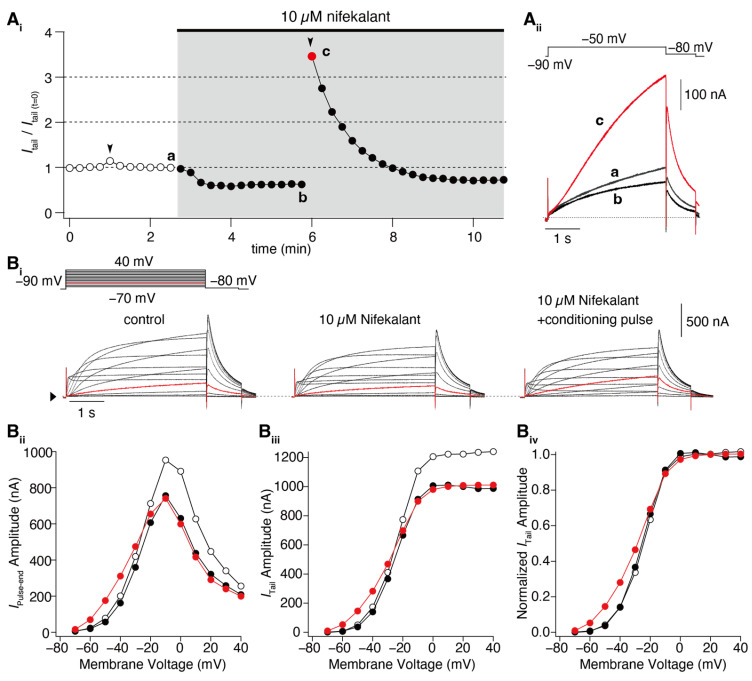
Nifekalant-induced facilitation of hERG activation. hERG channels were ectopically expressed in *Xenopus* oocytes, and the currents were recorded using a two-electrode voltage clamp method. (**A**) Depolarization-induced induction of facilitation. (**A_i_**) Time course of changes in hERG tail current (recorded at −80 mV) evoked by repetitive test pulses to −50 mV every 15 s. Conditioning pulses (+60 mV, 4 s) were applied twice in this experiment (black arrowheads), first in the absence and then in the presence of nifekalant. (**A_ii_**) Superimposed cell currents recorded in the same oocyte before (Time (a) in (**A_i_**)) and after perfusion with 10 µM nifekalant, with (c) or without a conditioning pulse (b). Increased hERG current after the induction of facilitation effect by nifekalant is highlighted with red (red circle and trace in (**A**_i_) and (**A**_ii_), respectively). (**B**) Nifekalant plus conditioning pulse induced a shift in hERG activation curves. (**B_i_**) Representative traces of hERG currents in the control (left), block (center), and block/facilitation (right) conditions. (**B_ii_**) IV relationship, (**B_iii_**) GV relationship, and (**B_iv_**) normalized GV relationship. Open, filled black, and filled red circles represent the control, 10 µM nifekalant without a conditioned pulse (block), and 10 µM nifekalant with conditioned pulse (block/facilitation) conditions, respectively. Panel (**A**) was adapted with permission from Ref. [37].

**Figure 2 ijms-24-16261-f002:**
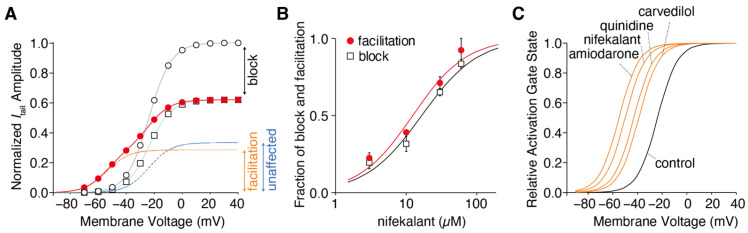
Drug-dependent negative shift in the hERG activation curve. (**A**) Voltage dependence of the hERG activation curves in the presence of 10 µM nifekalant. The tail currents of hERG in the absence (open circles) and presence of nifekalant with (filled circles) or without (open squares) the conditioning pulse were measured during the repolarizing pulse to −80 mV. The data were normalized to the current amplitude recorded following a voltage step of +10 mV in the absence of nifekalant. The model assumed two populations of channels with or without the facilitation effect of nifekalant (10 µM). The V_1/2_ of activation for the facilitated fraction (orange dashed line) of channel was −50.7 mV, almost 28 mV negative to that of control channel (blue dashed lines). The red lines represent the double Boltzmann function (the sum of the Boltzmann functions for the facilitated fraction (orange dashed line) and the unaffected fraction (blue dashed lines)). (**B**) Concentration–response relationships for compound-induced block and facilitation by nifekalant. (**C**) Drug-dependent GV relationship in the facilitated fraction of hERG channels. Panel A was adopted with permission from Ref. [37].

**Figure 3 ijms-24-16261-f003:**
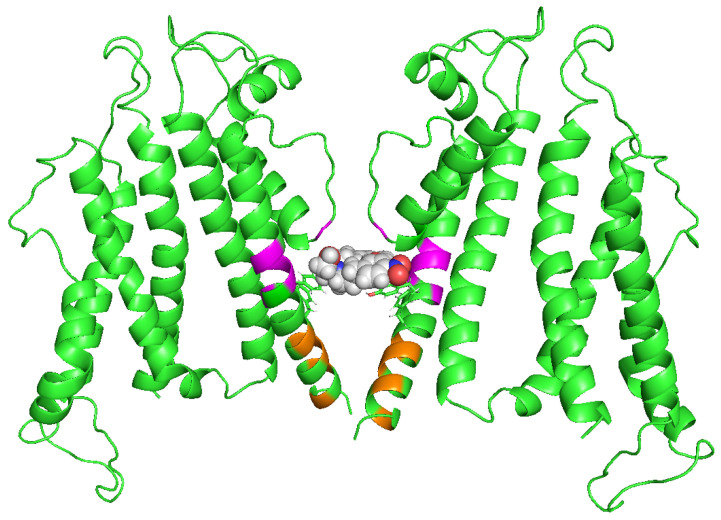
A possible hERG facilitator interaction site. Open hERG channel model (green). The side chains of Y652 and F656, crucial for the block, are shown as sticks. Magenta and orange indicate residues revealed experimentally by point mutagenesis influential for nifekalant’s facilitation but not block. Magenta residues (L646, I647, S649, and M651) form hydrophobic pockets. In contrast, orange residues (G657, V659, S660, I663, and R665) may be located on the activation gate. The drug (nifekalant) may be positioned deep within the hydrophobic pocket [68]. The image was kindly provided by Dr. Igor Vorobyov (University of California, Davis).

**Figure 4 ijms-24-16261-f004:**
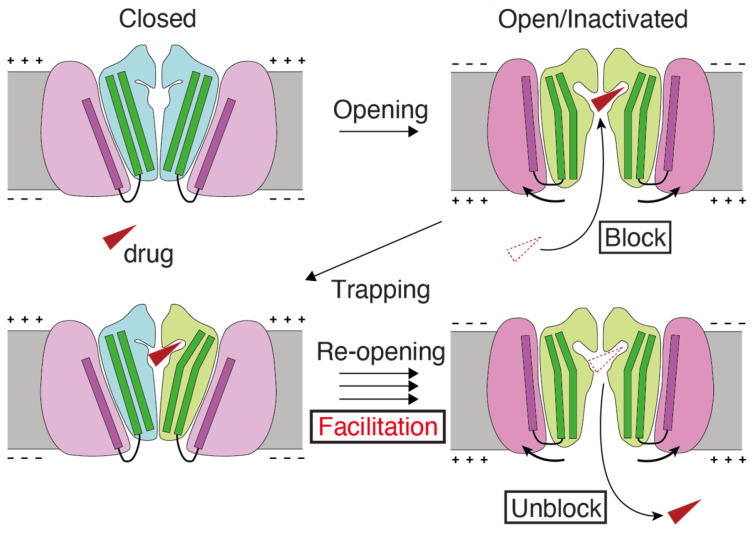
Schematic model of the mechanism underlying facilitation. (1) The hERG facilitator (red triangle) accesses the receptor site within the pore of the open or inactivated channels at depolarized potentials and “blocks” the channel. (2) Upon return to the resting potentials, channels close and trap the drug inside. (3) Trapped, the drug biases the open–closed equilibrium towards the open state and “facilitates”. (4) The drug can escape from the re-opened hERG channels, creating open channels. Drug translocations to block and unblock are from red dotted to solid triangles. The cartoon of structural changes during hERG channel gating draws on the graphical abstract of Wang and MacKinnon’s cell article [55].

**Figure 5 ijms-24-16261-f005:**
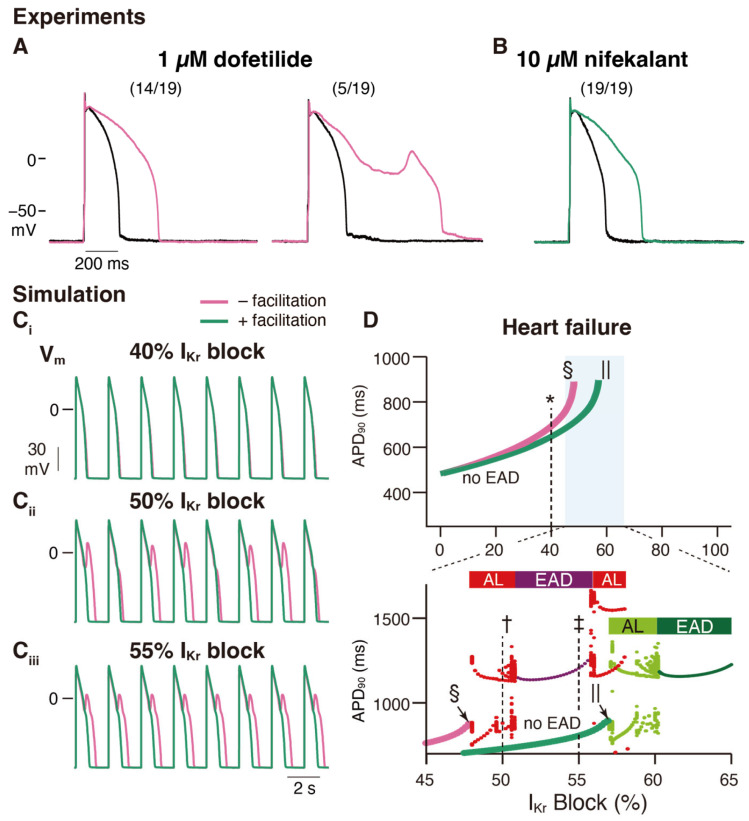
Facilitation suppresses the development of early afterdepolarization related to *I*_Kr_ blockade in ventricular cardiac myocytes. (**A**,**B**) Experimental study: Rabbit ventricular myocyte APs are more stable in nifekalant (**B**) than in dofetilide (**A**). AP responses in isolated rabbit ventricular myocytes were stimulated by minimal current injection (0.5 Hz) in whole-cell current clamp mode at 37 °C. Black line in (**A**,**B**) are control AP responses before the treatment. (**A**) Representative AP responses without (left) or with (right) EAD in 1 µM dofetilide. Of the 19 cells treated with 1 µM dofetilide (magenta), five showed EAD responses (26%). Fourteen cells showed prolonged APD at 1 µM but did not show EAD responses. (**B**) Representative AP responses to 10 µM nifekalant. All cells treated with 10 µM nifekalant (green) showed prolongation of APD upon treatment with 10 µM nifekalant but did not show EAD responses. (**C**,**D**) Simulation study: The effect of *I*_Kr_ facilitation on APD prolongation and EAD development by *I*_Kr_ block. (**C**) Steady-state AP trains with 40% (**C_i_**), 50% (**C_ii_**), and 55% (**C_iii_**) *I*_Kr_ block in a heart failure model with and without facilitation. (**D**) Effect of *I*_Kr_ blockade and facilitation on APD and development of EADs in the heart failure model. Green and magenta lines indicate APD90 of AP (without EAD) for block with and without facilitation, respectively. The asterisk, dagger, and double-dagger indicate the conditions in (**C**), respectively. The sections and pipes indicate the upper limits of the *I*_Kr_ block, where APs are normally terminated. When EAD was observed, it was classified as either alternating or periodic EAD. In the bottom panel of D, red and deep purple dots indicate APD90 of AP with EAD for the block without facilitation, while light and deep green dots indicate APD90 of AP with EAD for the block with facilitation. Horizontal bars above the dots indicate alternating EAD, AL, periodic EAD, or EAD. This figure has been adapted with permission from Ref. [44].

**Figure 6 ijms-24-16261-f006:**
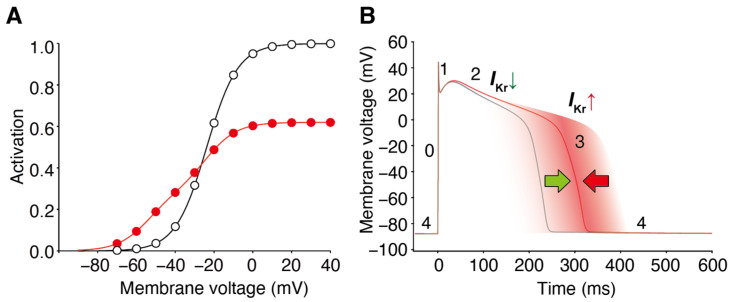
Block and facilitation impact on cardiac electrophysiology. (**A**) GV relationships of the control (open circles) and drug-treated conditions (red circles) intersect, resulting in drug-induced hERG/*I*_Kr_ currents decreasing at depolarized voltages and hERG/*I*_Kr_ currents increasing from the control at membrane voltages near the activation threshold. (**B**) The hERG channel blocker with a facilitation effect changes the attribute to *I*_Kr_ during the ventricular action potential. It decreases the current first and then increases it later, preventing excessive prolongation of the action potential and repolarization impairments. The different phases of the ventricular action potential (phases 0–4) are labeled. Green and red arrow indicate the dual drug actions on AP duration; *I*_Kr_ decrease by block prolongs (green), whereas *I*_Kr_ increase by facilitation prevents the prolongation of AP duration (red).

## Data Availability

The research data described in this paper are available upon request.

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
