# Peer review of "Facilitation of hERG Activation by Its Blocker: A Mechanism to Reduce Drug-Induced Proarrhythmic Risk"

_ijms, 2023, doi:10.3390/ijms242216261_

Round 1
Reviewer 1 Report
Comments and Suggestions for Authors
HERG channels are important for cardiac action potential repolarization. Dysfunctional hERG channels can lead to long QT syndrome and cardiac arrhythmias. In addition, they attract our attention as they can be blocked via off-target from non-cardiac drugs. The authors here reviewed the facilitation of hERG channels by some blockers and highlighted the potential pharmacological use for patients with cardiac arrhythmias. This review is thorough, organized and easy to understand. It would be exciting for a wide range of readers. However, there are some concerns that the author might need to address and clarify.
1. Line 165, what function was used to sum to the functions? Double Boltzmann?
2. The mechanism of D540K inducing facilitation needs more explanation. How does the D540K mutant data fit into the proposed model in Figure 4? I assume the voltage sensor moves similarly in wt and D540K channels.
3. Figure 3, the drug seems not to bind to the magenta areas supposed to be important for its facilitation. It would help a lot if the author could highlight some crucial residues, such as Y652 or F656, in the figure.
4. Is there a fenestration in hERG channels, would fenestration have something to do with the facilitation?
5. Missing legends in Figure 1B. What are the different colored circles?
6. In Figure 6, please label different phases in the AP in panel B, which helps to understand how the facilitation can be antiarrhythmic.
7. Is there facilitation in other repolarization-contributing Kv channels in the heart, such as IKs? This could be included in the manuscript.
Reviewer 2 Report
Comments and Suggestions for Authors
I have enjoyed reading your carefully put together and quite comprehensive review concerning the phenomenon that you describe as facilitation of hERG channel activation by certain chemical compounds including some drugs. The summarized information, based mainly on your previous publications has the potential to be useful to a broad range of basic and clinical cardiac electrophysiologists. However, before it is further considered for publication, it will be necessary for you to rewrite key sections and edit others. Perhaps the most direct and simple way for me to convey my requests is to provide for you my suggested edits. These appear in handwriting on the attached draft manuscript. In addition, key sections of the manuscript are unclear or raise mechanistic questions. Each of these must be addressed and each is denoted in the manuscript by the comments that are highlighted in yellow.
After reading your review I was also interested in a fundamental feature of your data analysis. It remains unclear to me what aspect of the ion/ion interaction within the pore or other alterations in the kinetics of the inactivation process actually produced the change in the activation threshold for the hERG current that your previous experimental data and modeling suggests is the mechanism for 'facilitation'. For example, do you view this as being completely independent of surface charge regulation of the gaiting of this channel? Secondly, how does this modulation of hERG channel gaiting relate to the known and marked alteration of hERG channel pharmacology by changes in plasma potassium levels? If new text concerning these issues could be added when you revise the manuscript, it is likely that the impact of this review will be enhanced.

Comments on the Quality of English LanguageAdequate but requires extensive editing.
Round 2
Reviewer 2 Report
Comments and Suggestions for Authors
Thank you for considering my suggestions for changes and clarifications and for providing answers to my general questions.